# Electrolyzed Hydrogen Water Alleviates Abdominal Pain through Suppression of Colonic Tissue Inflammation in a Rat Model of Inflammatory Bowel Disease

**DOI:** 10.3390/nu14214451

**Published:** 2022-10-22

**Authors:** Di Hu, Tianliang Huang, Mika Shigeta, Yuta Ochi, Shigeru Kabayama, Yasuyoshi Watanabe, Yilong Cui

**Affiliations:** 1Laboratory for Biofunction Dynamics Imaging, RIKEN Center for Biosystems Dynamics Research, Kobe 6500047, Japan; 2Trim Medical Institute, Co., Ltd., Osaka 5300001, Japan; 3Laboratory for Pathophysiological and Health Science, RIKEN Center for Biosystems Dynamics Research, Kobe 6500047, Japan

**Keywords:** electrolyzed hydrogen water, inflammatory bowel disease, 2,4,6-trinitrobenzene sulfonic acid, abdominal pain, reactive oxygen species

## Abstract

Inflammatory bowel disease (IBD) is characterized by chronic inflammation of the digestive tract and is typically accompanied by characteristic symptoms, such as abdominal pain, diarrhea, and bloody stool, severely deteriorating the quality of the patient’s life. Electrolyzed hydrogen water (EHW) has been shown to alleviate inflammation in several diseases, such as renal disease and polymyositis/dermatomyositis. To investigate whether and how daily EHW consumption alleviates abdominal pain, the most common symptom of IBD, we examined the antioxidative and anti-inflammatory effects of EHW in an IBD rat model, wherein colonic inflammation was induced by colorectal administration of 2,4,6-trinitrobenzene sulfonic acid (TNBS). We found that EHW significantly alleviated TNBS-induced abdominal pain and tissue inflammation. Moreover, the production of proinflammatory cytokines in inflamed colon tissue was also decreased significantly. Meanwhile, the overproduction of reactive oxygen species (ROS), which is intricately involved in intestinal inflammation, was significantly suppressed by EHW. Additionally, expression of S100A9, an inflammatory biomarker of IBD, was significantly suppressed by EHW. These results suggest that the EHW prevented the overproduction of ROS due to its powerful free-radical scavenging ability and blocked the crosstalk between oxidative stress and inflammation, thereby suppressing colonic inflammation and alleviating abdominal pain.

## 1. Introduction

Inflammatory bowel disease (IBD) comprises two major forms of chronic immune-mediated disorders of the gastrointestinal tract: Crohn’s disease (CD) and ulcerative colitis (UC). IBD is characterized by the repeated aggravation of symptoms, such as diarrhea, abdominal pain, rectal bleeding, and mucous stool, severely deteriorating the quality of the patient’s life [1,2,3,4]. Although the precise etiologic cause of IBD remains unclear, a malfunction of the intestinal immune system is believed to be critically related to IBD [5]. Recent preclinical and clinical studies have demonstrated that inflammatory mediators released from inflamed colon tissue might be involved in sensitizing afferent nerve terminals in the surrounding intestinal tract [6,7,8]. For instance, pro-inflammatory cytokines such as interleukin (IL)-1β and tumor necrosis factor (TNF)-α released from macrophages activate the arachidonic acid cascade in the fibroblasts and vascular endothelial cells to produce prostaglandin E2 (PGE2). PGE2 evokes hypersensitivity of C-fiber nociceptors in inflamed colon tissue [9,10,11]. Similarly, expression levels of transient receptor potential vanilloid 1 (TRPV1) and other members of the TRP family were reported to be higher in inflamed colon tissue [8,12,13]. The TRP family of receptors are the representative nociceptive sensors that play a critical role in peripheral hypersensitivity.

Anti-inflammatory drugs, such as amino salicylates, are primarily used to treat IBD, reduce symptoms, and maintain remission [14]. More recently, several therapeutic biological drugs have been developed, and more than a half dozen biologics, e.g., anti-TNF antibodies, have been approved for the treatment of human IBD [14,15]. These observations suggest that anti-inflammatory interventions could be an effective approach to alleviate abdominal pain in IBD.

Electrolyzed hydrogen water (EHW), an electrochemically reduced water enriched with molecular hydrogen, suppressed inflammatory processes in several chronic diseases [16,17,18]. For example, Yoon et al. [18] reported that EHW suppressed inflammatory responses and alleviated atopic dermatitis in an animal model. The anti-inflammatory effect of EHW is believed to be due to its superior antioxidative effect, as molecular hydrogen is a potent free-radical scavenger. For instance, consuming water containing molecular hydrogen suppressed oxidative stress in various inflammatory animal models, resulting in the downregulation of the nuclear factor-kappa B (NF-κB) signaling cascade, and preventing the increase in pro-inflammatory cytokines, such as IL-1β and TNF-α expression [18,19,20,21]. These potentially beneficial effects of EHW in chronic diseases have been recognized nationwide in Japan. Various types of hydrogen water including the EHW are commercially available for daily consumption. Therefore, daily EHW consumption has been expected to be an effective approach for controlling chronic diseases, such as IBD, characterized by repeated remission and relapse. However, whether and how daily EHW consumption alleviates abdominal pain in IBD has remained unclear.

Abdominal pain is the most common symptom in patients with IBD and is considered the major therapeutic target. Clinical and preclinical studies have demonstrated that abdominal pain can be objectively assessed by visceromotor response (VMR) to colorectal distension (CRD) based on the electromyogram (EMG) activity in the external oblique muscle [22,23]. Recently, we developed an effective method for quantitatively evaluating VMR in conscious rats by chronic implantation of EMG recording electrodes into the external oblique muscle [22]. In the present study, we induced IBD in rats by colorectal administration of ethanol and 2,4,6-trinitrobenzene sulfonic acid (TNBS), a chemical compound widely used to induce IBD in animals. In this model, ethanol effectively removes the intestinal barrier, and TNBS directly evoke inflammatory pathways to generate experimental colitis in rats [24,25]. Here, we investigated whether and how EHW consumption reduces abdominal pain following persistent colonic inflammation induced by TNBS.

## 2. Materials and Methods

### 2.1. Experimental Animals

Eight-week-old male Wistar rats (SLC Inc., Hamamatsu, Japan) were used in this study. The rats were housed under a 12 h light/dark cycle condition at 23.0 ± 0.5 °C and fed food and water ad libitum. All experimental protocols were approved by the Institutional Animal Care and Use Committee of RIKEN, Kobe Branch, and all experiments were performed in accordance with the principles of laboratory animal care (8th edition, National Academies Press, 2011). This study was conducted in compliance with the Animal Research: Reporting in Vivo Experiments (ARRIVE) guidelines [26]. Furthermore, all possible means were taken to minimize animal suffering.

### 2.2. IBD Model

The IBD rat model was generated by colorectal TNBS administration (Nacalai Tesque, Kyoto, Japan), as reported previously [22,24]. Prior to TNBS administration, rats were fasted for 24 h, and then, TNBS dissolved in 50% ethanol (final concentration, 40 mg/mL; dose, 0.625–1.25 mL/kg) was administered into the rat colon, 10 cm proximal from the anus, using a syringe needle, under 1.5% isoflurane anesthesia, and the rats were maintained in a head-down position for 30 min.

### 2.3. Implantation of the Electrodes

One week before the experiment, three Teflon-insulated flexible wires (AS 633; Cooner Wire, Chatsworth, CA, USA) were implanted into the external oblique muscle of rats to record abdominal muscle contraction. The animals were anesthetized with a mixture of 1.5% isoflurane and oxygen/nitrous oxide (3:7). A flexible wire was sutured into the left external oblique muscle. The other flexible wires were sutured into the right external oblique muscle at a distance of approximately 1.5 cm. All electrode leads were threaded through the subcutaneous tissue, externalized at the back of the rats, and anchored to the back skin with a suture. During surgery, the animal’s body temperature was maintained at approximately 37 °C using a heating pad. Rats were allowed to recover from surgery for 5 days prior to the experiment.

### 2.4. Visceromotor Response to Colorectal Distension

The rats were allowed to sufficiently habituate to a cylindrical box (diameter: 6 cm, length: 16 cm) and an uninflated balloon was inserted into the colon for 15 min every 2 days. Visceromotor response (VMR) to colorectal distension (CRD) was measured after a few days of habituation to evaluate the severity of abdominal pain in conscious rats. Rats were adequately habituated and remained quiet even during CRD, as reported previously [22,23]. Briefly, rats were left conscious in a cylindrical box (diameter, 6 cm; length, 16 cm), and a balloon (length: 4 cm) was inserted 10 cm proximal to the anus. The balloon catheter was fixed to the rat’s tail with tape. After 5 min of habituation, CRD was performed twice for 10 s with a 5 min interval. EMG signals were recorded from the external oblique muscles. After filtering (0.3–1000 Hz) and amplifying the EMG signal using a biological amplifier (Bio Amp; AD Instruments, Sydney, Australia), they were saved using a PowerLab system (PowerLab 8/35, Chart, AD Instruments, Sydney, Australia). The VMR threshold was defined as the pressure at which the EMG activity increased to 5-fold higher than the value at the baseline (10 s) before the CRD.

### 2.5. EHW Treatment

EHW was produced using a commercial generator (Trim Ion Grace, Nihon Trim Co. Ltd., Osaka, Japan), and connected with a drinking nozzle inserted into each cage to allow rats to drink freely. EHW was refreshed every 30 min during the experiment. The concentration of dissolved hydrogen in EHW from the drinking nozzle was measured using a flow cell-type hydrogen sensor (DH-35A, TOADKK, Tokyo, Japan) that ranged between 600–700 ppb. In order to examine the effects of daily EHW consumption, EHW administration was started 10 days prior to TNBS treatment, and lasted until the end of the experiment. In the control group, tap water was administered for the same duration.

### 2.6. Measurement of Myeloperoxidase (MPO) and S100A9 Levels

Blood samples were collected from the tail vein of rats before and after TNBS treatment using a heparin-treated collection tube while the animals were anesthetized with a mixture of 1.5% isoflurane and oxygen/nitrous oxide (3:7). The plasma was separated by centrifugation at 12,000 rpm for 10 min at 4 °C. At the end of the experiment, each rat was deeply anesthetized and perfused transcardially with 30 mL of 0.1 M phosphate-buffered saline (PBS, pH 7.4). Colon tissue was removed and homogenized in 10 volumes of ice-cold 200 mM sodium phosphate buffer (pH 7.4) containing 5 mM EDTA, 1 mM PMSF, 1 µg/mL leupeptin, and 28 µg/mL aprotinin. The homogenate was centrifuged at 12,000 rpm for 10 min at 4 °C. The concentration of S100A9 (Rat S100A9 ELISA kit, Yamasa Corporation, Chiba, Japan) in the plasma and MPO (Rat MPO ELISA kit, Wayne, USA) in the colon tissue were measured following the manufacturers’ instructions.

### 2.7. Oxidative Stress Assay

To assess oxidative stress induced by the TNBS treatment, the superoxide dismutase (SOD) activity and reactive oxygen metabolites-derived compounds (d-ROMs) were measured using the commercial kits, SOD assay kit-WST (Dojindo, Kumamoto, Japan) and d-ROMs kit (Wismerll, Tokyo, Japan), respectively, as described previously [27]. The d-ROMs kit was used to measure the total hydroperoxide in the plasma sample. The d-ROMs levels are expressed in arbitrary units called “Carratelli units” (U.CARR).

### 2.8. Real-Time PCR

Total RNA was extracted from the colon tissue using Sepasol-RNA I Super G (Nacalai Tesque, Kyoto, Japan) and purified using a column kit (Takara Bio, Shiga, Japan). The total RNA was quantified using a NanoDrop (Thermo Fisher Scientific, Waltham, MA, USA), and the concentration was adjusted to 50 ng/mL. cDNA was synthesized using the ReverTra Ace qPCR RT kit (TOYOBO, Osaka, Japan) following the manufacturer’s instructions. Real-time PCR was performed using the QuantStudio5 system (Thermo Fisher Scientific, Waltham, MA, USA) with KOD SYBR qPCR Mix (TOYOBO, Osaka, Japan). The specific primer sets were as follows: GAPDH, forward: 5′- GGCAAGTTCAATGGCACAGT -3′ and reverse: 5′- TGGTGAAGACGCCAGTAGACTC -3′; IL-1β, forward: 5′- GCAACTGTCCCTGAACTCAACT -3′ and reverse: 5′- ATCTTTTGGGGTCTGTCAGCC -3′; TNF-α, forward: 5′- CCCGTAGCCCACGTCGTAG -3′ and reverse: 5′- GGGAGTAGATAAGGTACAGCCC -3′; IL-6, forward: 5′- TAGTCCTTCCTACCCCAACTTCC -3′ and reverse: 5′- TTGGTCCTTAGCCACTCCTTC -3′; MCP-1, forward: 5′- GTGCTGACCCCAATAAGGAA -3′ and reverse: 5′- TGAGGTGGTTGTGGAAAAGA -3′; S100A9, forward: 5′- GTACTCTAGGAAGTATGGACATC -3′ and reverse: 5′- TGATTGTCCTGGTTTGTGTCC -3′.

### 2.9. Histological Analysis

At the end of the experiments, rats were deeply anesthetized with 5% isoflurane and perfused transcardially with 30 mL 0.1 M phosphate buffer (pH 7.4). Colon tissues were removed, cut into 1.5 cm sections, and fixed for 24 h using 4% paraformaldehyde in 0.1 M PBS. The tissues were transferred to a 30% sucrose solution for 72 h and stored at 4 °C. The tissue coronal sections were cut into 5 μm section using a cryostat, mounted onto glass slides, and stained with hematoxylin and eosin to examine tissue damage.

### 2.10. Statistical Analysis

Statistical analyses were performed using Prism 8.0 (GraphPad, San Diego, CA, USA). One-way analysis of variance (ANOVA) followed by Tukey′s multiple-comparison test was used to test the statistical differences in VMR, MPO, IL-1β, TNF-α, IL-6, MCP-1, S100A9, SOD, and d-ROMs levels before and after TNBS treatment. The statistical differences between the tap water- and EHW-administered groups were tested by two-way ANOVA followed by Bonferroni′s multiple-comparison test. Statistical significance was set at *p* < 0.05. Data are reported as mean ± standard error of the mean.

## 3. Results

### 3.1. Alleviation of TNBS-Induced Abdominal Pain in Rats by Consumption of EHW

Recently, we reported that the VMR threshold, the CRD pressure inducing a particular increment in the baseline EMG activity, is a sensitive biomarker for evaluating the severity of abdominal pain [20]. To investigate whether EHW consumption relieves TNBS-induced abdominal pain, we defined the VMR threshold as the pressure-inducing increment of more than five times the baseline EMG activity. We analyzed the VMR threshold in tap water- and EHW-administered groups before and up to 14 days after TNBS treatment. Prior to TNBS treatment, the VMR threshold in the tap water- and EHW-administered groups was 40.61 ± 2.93 mmHg and 41.00 ± 2.67 mmHg, respectively (Figure 1). Two days post TNBS treatment, the VMR threshold in the tap water group significantly decreased to 21.76 ± 2.34 mmHg (*p* = 0.0001) and gradually recovered (4 days, *p* = 0.0057; 6 days, *p* = 0.0130; 8 days, *p* = 0.0158) to pre-treatment level within a few weeks, as reported previously (one-way ANOVA followed by Tukey’s multiple comparison procedure) [22]. Similarly, the VMR threshold in the EHW-administered group significantly decreased to 26.80 ± 1.85 mmHg (*p* = 0.0008) and recovered subsequently (one-way ANOVA followed by Tukey’s multiple comparison procedure). However, the decrease in the VMR threshold was attenuated by EHW intake throughout the experimental period. In the EHW group, the reduction in the VMR threshold was significantly inhibited 6 (*p* = 0.0416) and 8 (*p* = 0.0002) days post TNBS treatment (two-way ANOVA followed by Bonferroni’s multiple comparison procedure). 

### 3.2. EHW Suppressed Colonic Tissue Inflammation

Inflammatory stimulation triggers hypersensitivity of afferent nerve terminals in inflamed colonic tissue [6,8]. To investigate whether the EHW-induced alleviation of abdominal pain resulted from its anti-inflammatory effect, we compared TNBS-induced tissue inflammation between the tap water- and EHW-administered groups. Histological analysis revealed severe tissue damage in the mucosa, submucosa, and muscular layer of the colon. We further observed inflammatory cell infiltration in the submucosa and circuit muscular layer 2 days post TNBS treatment in the tap water- and the EHW-administered groups (Figure 2). However, this inflammatory tissue damage was partially recovered in the EHW-administered group but not in the tap water group 8 days post TNBS treatment. In addition, a quantitative assay for MPO, the most abundant protein in neutrophils and a widely used biomarker for quantifying tissue inflammation, was also performed on inflamed colon tissue treated with TNBS. MPO levels were significantly increased in the tap water-administered group 2 (*p* = 0.0143) and 4 (*p* = 0.0218) days post TNBS treatment (one-way ANOVA followed by Tukey’s multiple comparison procedure) (Figure 3). Meanwhile, the increased MPO level in the EHW-administered group significantly reduced 2 (*p* = 0.0011) and 4 (*p* = 0.0009) days post TNBS treatment (two-way ANOVA followed by Bonferroni’s multiple comparison procedure).

### 3.3. EHW Inhibited TNBS-Induced Cytokine Expression in Inflamed Colon Tissue

Intestinal immune responses induced by TNBS treatment include infiltration of immune cells and a marked increase in cytokines associated with the progression of colitis [28,29,30]. To further evaluate the effect of EHW on the TNBS-induced immune response, we measured the expression levels of representative inflammatory mediators in inflamed colon tissue using real-time PCR. As shown in Figure 4, inflammatory cytokines IL-1β (*p* = 0.0399), TNF-α (*p* = 0.0152), IL-6 (*p* = 0.0125), and chemokine MCP-1 significantly increased during the initial period and gradually returned to pre-treatment levels within a few weeks (one-way ANOVA followed by Tukey’s multiple comparison procedure), as reported previously [25,26]. In the EHW-administered group, the increase in IL-6 (*p* = 0.0099) and MCP-1 (*p* = 0.0272) levels were significantly suppressed 2 days post the TNBS treatment. IL-1β (*p* = 0.0003), TNF-α (*p* = 0.0003), and IL-6 (*p* = 0.0025) levels were suppressed up to 4 days post TNBS treatment (two-way ANOVA followed by Bonferroni’s multiple comparison procedure). These results indicated that EHW consumption relieved the intestinal immune responses induced by the TNBS treatment. 

### 3.4. EHW Suppressed TNBS-Induced Systemic Oxidative Stress

ROS overproduction plays an essential role in IBD pathogenesis and is closely associated with the development of intestinal inflammation [31]. To confirm whether and how the redox state is altered by TNBS treatment and further affected by EHW consumption, we assessed the total superoxide dismutase (SOD) activity in inflamed colon tissue. As reported previously [32,33,34], total SOD activity was decreased in inflamed colon tissue 2, 4 and 8 days post the TNBS treatment in the tap water-administered group (Figure 5A). However, the decrease in the total SOD activity was counteracted by EHW consumption. Significant differences between tap water- and EHW-administered groups were observed 2 (*p* = 0.0065) and 8 (*p* = 0.0018) days post the TNBS treatment (two-way ANOVA followed by Bonferroni’s multiple comparison procedure). 

In addition, we measured the plasma levels of ROS using the reactive oxygen metabolites-derived compounds (d-ROMs) assay. The d-ROMs assay has been widely used to measure ROS-derived total hydroperoxide. Hydroperoxide generates alkoxyl and peroxyl radicals in the presence of iron; hydroperoxide oxidizes and transforms its substrate into a pink-colored stable derivative that can be quantified photometrical [27]. The plasma d-ROMs levels were significantly elevated 2 (*p* = 0.0002) and 4 (*p* = 0.0232) days post the TNBS treatment in the tap water-administered group (one-way ANOVA followed by Tukey’s multiple comparison procedure) (Figure 5B). Conversely, the elevated plasma d-ROMs levels were significantly suppressed by EHW-administration 2 (*p* = 0.0182) and 4 (*p* = 0.0269) days post the TNBS treatment (two-way ANOVA followed by Bonferroni’s multiple comparison procedure). 

### 3.5. EHW Reduced Circulating S100A9 Level

Finally, we examined whether the level of S100A9, a clinical biomarker for IBD, was altered along with tissue inflammation in our experimental setting. Fecal calprotectin, a heterodimeric complex of the S100A8 and S100A9 proteins, has been approved by the FDA as an intestinal inflammation biomarker for diagnosing and predicting disease activity in patients with IBD [35,36]. Circulating levels of S100A9 also correlate strongly with the histological score of disease severity of ulcerative colitis [36]. As expected, the plasma level of S100A9 was significantly increased in the tap water-administered group 2 (*p* = 0.0037) days post the TNBS treatment (one-way ANOVA followed by Tukey’s multiple comparison procedure) when the most severe intestinal inflammation and lowest VMR threshold were observed (Figure 6). Meanwhile, the increase in plasma S100A9 level was significantly reduced in the EHW-administered group (*p* = 0.0071, two-way ANOVA followed by Bonferroni’s multiple comparison procedure), further supporting the anti-inflammatory effect of EHW in IBD pathophysiology.

## 4. Discussion

In the present study, we examined whether and how EHW alleviates abdominal pain associated with persistent colonic inflammation. We used an IBD rat model to demonstrate, for the first time, that EHW consumption suppressed TNBS-induced inflammatory responses and oxidative stress, thereby alleviating abdominal pain in IBD. We provide new lines of evidence that EHW consumption: (1) significantly suppressed an increase in the inflammatory response, including cytokine levels and inflammatory cell infiltration induced by TNBS treatment; (2) suppressed the elevation of oxidative stress following persistent colitis; (3) reduced the increased level of IBD clinical biomarker S100A9; and (4) alleviated abdominal pain.

The main finding of this present study is that EHW consumption alleviated abdominal pain induced by persistent colonic inflammation in an IBD rat model. Recent reports have provided lines of evidence that malfunction of the immune system in the digestive tract may be a crucial mechanism for the onset and progression of IBD [2,5]. Clinical and preclinical studies on IBD pathophysiology demonstrated that multiple inflammatory mediators released from inflamed colon tissue directly contribute to the induction of diverse symptoms of IBD, including abdominal pain [6,7,8]. For instance, the increased IL-1β and TNF-α in inflamed tissue stimulate surrounding fibroblasts and vascular endothelial cells to induce COX-2 upregulation, leading to the generation of PGE2. PGE2 binds to the EP1/EP2 receptor expressed on C-fiber nociceptors to evoke visceral hypersensitivity [9,10,11]. Consistent with these observations, we found that inflammatory mediators, such as IL-1β, IL-6, TNF-α, and MCP-1, were significantly increased in inflamed colon tissue. The visceral pain threshold in response to CRD was significantly decreased after TNBS treatment (Figure 1 and Figure 3). These results suggest that the increased levels of inflammatory mediators in inflamed colon tissue induce hypersensitivity of peripheral nociceptors and decrease the CRD-induced VMR threshold after TNBS treatment in the tap water group. Meanwhile, the anti-inflammatory effect of EHW has been revealed by clinical and preclinical studies in various diseases, such as renal disease and polymyositis/dermatomyositis [37,38]. The studies showed that EHW prevented an increase in pro-inflammatory cytokines, such as IL-1β and TNF-α expression, via suppression of NF-κB pathway in various inflammatory animal models [18,19,20,21]. Accordingly, our data showed that EHW consumption suppressed tissue inflammation and mitigated the increased expression of inflammatory mediators, such as IL-1β, IL-6, TNF-α, and MCP-1, in inflamed colon tissues, post the TNBS treatment. In addition to inducing expression of nociceptor sensitizer such as PGE2, IL-1β is also known to directly sensitized the peripheral terminals of sensory neurons to induce pain hypersensitivity via second messenger actions (e.g., PKA, PKC, and p38 MAPK) [12,39]. 

In contrast, blocking the signaling cascade using neutralizing antibodies or IL-1R antagonist suppresses hyperalgesia in colitis model [40,41]. Moreover, chemokines such as MCP-1, bind to chemokine receptor CCR2 in peripheral neurons, mediate pain hypersensitivity via activation of TRPV-1, and directly excite primary nociceptive neurons within sacral dorsal root ganglia (DRG) [42]. In contrast, CCR2 receptor antagonists inhibit the downstream of MCP-1/CCR2 signaling and reduce visceral hypersensitivity [43]. Consistent with these observations, we demonstrated that the TNBS-induced decrease in the VMR threshold in response to CRD in inflamed colon tissue was significantly alleviated by EHW consumption. These results suggest that EHW consumption reduced the production of inflammatory mediators in inflamed colon tissue, thereby preventing the development of peripheral nociceptor hypersensitivity and alleviating abdominal pain. 

ROS overproduction plays an essential role in IBD pathophysiology [31]. Clinical studies have reported significantly decreased SOD activity and increased ROS production in colorectal biopsy specimens obtained from IBD patients [32,33,34]. Consistent with these reports, we observed decreased SOD activity in inflamed colon tissue and increased d-ROMs levels in plasma post TNBS treatment (Figure 5). Several reports demonstrated that ROS and their reactive products mediated persistent pain, including inflammatory pain by various mechanisms [44,45]. ROS scavengers, such as phenyl-N-t-butyl nitrone (PBN), 5,5-dimethylpyroline N-oxide (DMPO), and vitamin E, have been reported to alleviate pain responses [46]. For instance, systemic PBN injection significantly inhibit TRPV-1 expression via TNF receptor type 1 (TNFR1) and prevent hyperalgesia [45]. Moreover, increased ROS and their reactive products upregulate inflammatory mediator expression via activation of NF-κB signaling pathway [29,30]. Concurrently, increased inflammatory mediators can active NADPH-Oxygenase (NOX) expression that facilitate ROS from NADPH due to the respiratory bursts in inflammatory cells triggered in inflamed tissue [47]. This type of complicated crosstalk between ROS and inflammatory mediators forms a positive feedback loop and may contribute to IBD development [31,48,49]. EHW is enriched with molecular hydrogen, which selectively scavenges active oxygen species and free radicals such as hydroxyl radicals (•OH)/peroxynitrite (ONOO^−^) radicals, and has been proposed as an effective treatment for various diseases due to its antioxidant effects [50]. For example, a study based on a gastric injury model showed that molecular hydrogen showed powerful free-radical scavenging properties to remove cytotoxic ROS and rapidly diffused across membranes to protect stomach tissue from aspirin-induced inflammatory injury [17]. Similarly, our data showed that EHW consumption significantly suppressed the increase in d-ROMs and recovered the SOD activity in inflamed colon tissue post TNBS treatment. Overall, our results indicate that EHW suppressed ROS overproduction and blocked the crosstalk between ROS and inflammatory mediators to restrain the pathophysiology of IBD, consequently alleviating abdominal pain.

Finally, we confirmed that S100A9 level, an IBD biomarker, significantly increased post TNBS treatment, this increase was suppressed by EHW consumption (Figure 6). S100A9 is a calcium-binding protein overexpressed in immune cells of myeloid origin, such as neutrophils, monocytes, and dendritic cells, during inflammation. Immune cells infiltrating inflamed intestinal tissue further increase S100A9 levels [51,52]. In addition, S100A9 increases the expression of cytokines via NF-κB and AP-1 signaling pathways in a process intimately linked to ROS generation, which is involved in autophagy and apoptosis [53,54,55]. EHW consumption reduced S100A9 levels, mitigated the increase in pro-inflammatory cytokine levels, and inhibited ROS overproduction induced by S100A9. Therefore, S100A9 can also be used as a biomarker to monitor inflammation severity and predict the course of IBD, particularly playing a role in differentiating IBD from functional gut disorders [35,36,56]. These observations further support our finding that EHW consumption could suppress inflammatory responses and oxidative stress in inflamed colonic tissue and alleviate abdominal pain in IBD.

This study has several limitations that should be considered. First, to investigate how daily EHW consumption alleviates abdominal pain in the IBD rat model, EHW administration was started 10 days prior to the TNBS treatment. Due to this pretreatment regimen, precise therapeutic efficacy was difficult to predict. Future studies should examine the therapeutic effect at the time of administration of EHW post TNBS treatment. Second, although the alleviating effect of EHW in abdominal pain was demonstrated to be due to the antioxidative and anti-inflammatory effects of EHW, more detailed investigations are needed to confirm whether and how EHW affects peripheral sensory nerve sensitization. Moreover, based on the promising data in this study, we plan to use EHW consumption or colon hydrotherapy with EHW in the clinic for the patients with IBD, especially with ulcerative colitis. We expect the beneficial effects to be visible in a stratified clinical study with mild-moderate cases.

## 5. Conclusions

We demonstrated that EHW consumption alleviated abdominal pain associated with IBD and suppressed the elevation of inflammatory mediators and oxidative stress, as well as TNBS-induced intestinal tissue damage, possibly disrupting the crosstalk between increased ROS and inflammatory mediators.

## Figures and Tables

**Figure 1 nutrients-14-04451-f001:**
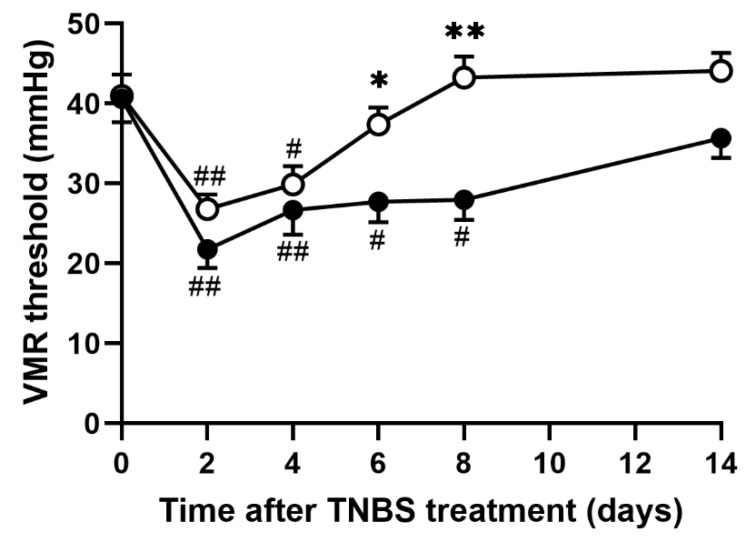
VMR in response to CRD in TNBS-treated tap water- and EHW-administered rats. VMR threshold level (mmHg) was detected before (pre) and 2, 4, 6, 8, and 14 days after TNBS treatment (tap water-administered, closed circle, n = 11; EHW-administered, open circle, n = 11). Each value represents the mean ± standard error of the mean. ^#^ *p* < 0.05, ^##^ *p* < 0.01 versus Pre TNBS-treatment. One-way ANOVA followed by Tukey’s multiple-comparison test. * *p* < 0.05, ** *p* < 0.01 significant difference between tap water- and EHW-administered groups. Two-way ANOVA followed by Bonferroni’s multiple-comparison test. VMR, visceromotor response; CRD, colorectal distension; EHW, electrolyzed hydrogen water; TNBS, 2,4,6-trinitrobenzene sulfonic acid.

**Figure 2 nutrients-14-04451-f002:**
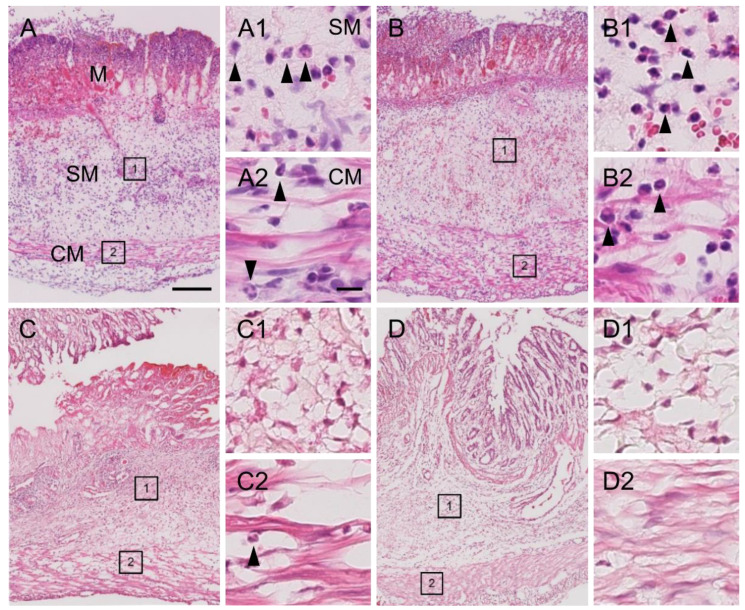
Photomicrographs of colon tissue obtained from TNBS-treated tap water- and EHW-administered rats. Representative light photomicrographs of colon tissue sections stained with hematoxylin-eosin showing tissue damage and inflammatory cell infiltration 2 (**A**,**B**) and 8 (**C**,**D**) days post TNBS treatment (tap water-administered, (**A**,**C**); EHW-administered, (**B**,**D**)). Enlarged areas of SM (1) and CM (2) in **A**–**D** are shown in high-magnification photomicrographs on the right side of each figure (**A**1 and **A**2, **B**1 and **B**2, **C**1 and **C**2, **D**1 and **D**2, respectively). Arrowheads in **A**1, **A**2, **B**1, **B**2, and **C**2 indicate representative inflammatory cells. Scale bars in (**A**) and (**A**2) indicate 200 µm and 50 µm, respectively. M, mucosa; SM, submucosa; CM, circuit muscle.

**Figure 3 nutrients-14-04451-f003:**
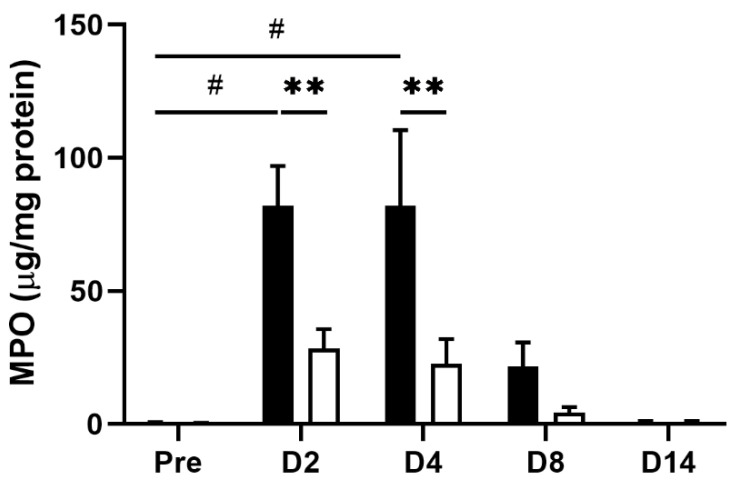
MPO level in colon tissue obtained from TNBS-treated tap water- and EHW-administered rats. MPO levels in colon tissue were detected before (pre) (tap water-administered, black bar, n = 4; EHW-administered, white bar, n = 5) and 2, 4, 8, and 14 days post TNBS treatment (tap water-administered, black bars, n = 7–8; EHW-administered, white bars, n = 7–10). The concentration of MPO (µg/mL) in colon tissue was normalized by total protein (mg/mL). Each value represents the mean ± standard error of the mean. ^#^ *p* < 0.05, versus. Pre TNBS-treatment. One-way ANOVA followed by Tukey’s multiple-comparison test. ** *p* < 0.01 significant difference between tap water- and EHW-administered groups. Two-way ANOVA followed by Bonferroni’s multiple-comparison test. MPO, myeloperoxidase.

**Figure 4 nutrients-14-04451-f004:**
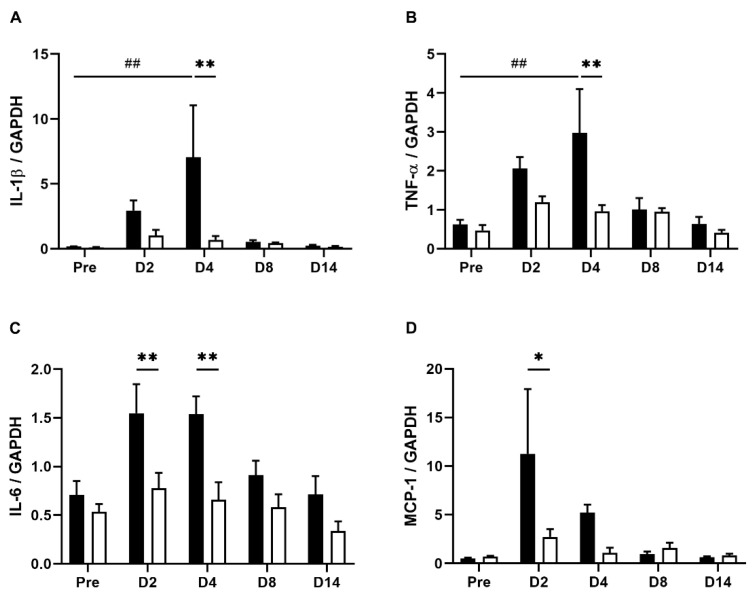
Levels of IL-1β, TNF-α, IL-6, and MCP-1 in the colon tissues of TNBS-treated tap water- and EHW-administered rats. Expression of IL-1β (**A**), TNFα (**B**), IL-6 (**C**), and MCP-1 (**D**) in inflamed colon tissues were quantified by real-time PCR before (pre) (tap water-administered, black bar, n = 4; EHW-administered, white bar, n = 5) and 2, 4, 8, and 14 days post the TNBS treatment (tap water-administered, black bars, n = 7–8; EHW-administered, white bars, n = 7–10). Expression of IL-1β, TNFα, IL-6 and MCP-1 was normalized to GAPDH expression. Each value represents the mean ± standard error of the mean. ^##^ *p* < 0.01 versus Pre TNBS-treatment. One-way ANOVA followed by Tukey’s multiple-comparison test. * *p* < 0.05, ** *p* < 0.01 significant difference between tap water- and EHW-administered groups. Two-way ANOVA followed by Bonferroni’s multiple-comparison test. IL, interleukin; TNF, tumor necrosis factor; MCP, monocyte chemoattractant protein.

**Figure 5 nutrients-14-04451-f005:**
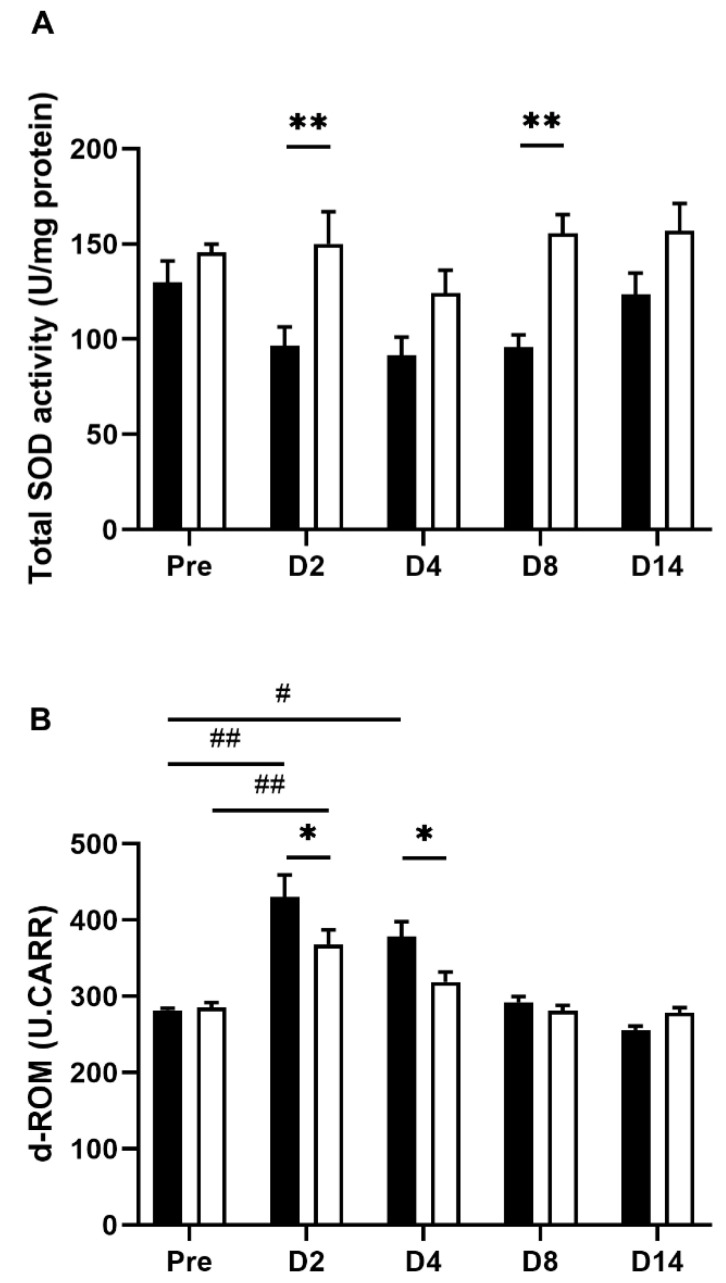
Oxidative stress in TNBS-treated tap water- and EHW-administered rats. (**A**) Total SOD activity (U/mg protein) in inflamed colon tissue was detected before (pre) (tap water-administered, black bar, n = 4; EHW-administered, white bar, n = 5) and 2, 4, 8, and 14 days post the TNBS treatment (tap water-administered, black bars, n = 7–8; EHW-administered, white bars, n = 7–10). (**B**) d-ROMs level [Carratelli units (U.CARR)] in the plasma was detected before (pre) (tap water-administered, black bar, n = 4; EHW-administered, white bar, n = 5) and 2, 4, 8, and 14 days post the TNBS treatment (tap water-administered, black bars, n = 7–8; EHW-administered, white bars, n = 7–10). Each value represents the mean ± standard error of the mean. ^#^ *p* < 0.05, ^##^ *p* < 0.01 versus Pre TNBS-treatment. One-way ANOVA followed by Tukey’s multiple-comparison test. * *p* < 0.05, ** *p* < 0.01, significant difference between tap water- and EHW-administered groups. Two-way ANOVA followed by Bonferroni’s multiple-comparison test. SOD, superoxide dismutase; d-ROMs, reactive oxygen metabolites-derived compounds.

**Figure 6 nutrients-14-04451-f006:**
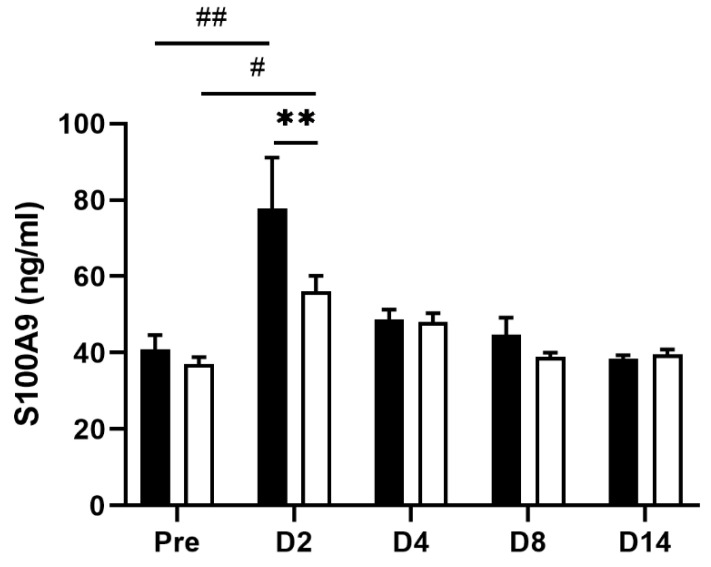
Change of plasma levels of S100A9 in TNBS-treated tap water- and EHW-administered rats. S100A9 levels (ng/mL) in plasma were detected before (pre) (tap water-administered, black bar, n = 4; EHW-administered, white bar, n = 5) and 2, 4, 8, and 14 days post the TNBS treatment (tap water-administered, black bars, n = 7–8; EHW-administered, white bars, n = 7–10). Each value represents the mean ± standard error of the mean. ^#^ *p* < 0.05, ^##^ *p* < 0.01 versus Pre TNBS-treatment. One-way ANOVA followed by Tukey’s multiple-comparison test. ** *p* < 0.01 significant difference between tap water- and EHW-administered groups. Two-way ANOVA followed by Bonferroni’s multiple-comparison test. S100A9, S100 calcium-binding protein A9.

## Data Availability

The data that support the findings of this study are available from the corresponding author upon reasonable request.

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
