# Peer review of "Electrolyzed Hydrogen Water Alleviates Abdominal Pain through Suppression of Colonic Tissue Inflammation in a Rat Model of Inflammatory Bowel Disease"

_nutrients, 2022, doi:10.3390/nu14214451_

Round 1

Reviewer 1 Report

This is a well-designed experimental study with translational significance. However there are a few minor issues need to be carefully addressed. 

1. In Discussion, please provide limitation of the present study. For example, (1) the pretreatment suggest the anti-inflammatory action, but its therapeutic potential is not tested yet; (2) the underlying molecular mechanisms need to be future investigated. 

2. In Discussion, please provide a brief statement on the future research direction including clinical translational research.   

Author Response

Reviewer 1 comments

This is a well-designed experimental study with translational significance. However there are a few minor issues need to be carefully addressed. 

  1. In Discussion, please provide limitation of the present study. For example, (1) the pretreatment suggest the anti-inflammatory action, but its therapeutic potential is not tested yet; (2) the underlying molecular mechanisms need to be future investigated.

>>>Responses:

We greatly appreciate the reviewer’s comments that allowed us to improve the quality of the manuscript.  According to the reviewer’s comments, we added a paragraph regarding the limitations at the end of the Discussion section. (Page 12, Line 432-441. In red letters)

  1. In Discussion, please provide a brief statement on the future research direction including clinical translational research.

>>>Responses:

We greatly appreciate the reviewer’s comments.  We added a future research direction of EHW in the Discussion section. (Page 12, Line 441-445. In red letters)

Reviewer 2 Report

In general, the study was well-performed and manuscript is well-written. Please explain why the TNBS model was used for the study.

Author Response

Reviewer 2 comments

In general, the study was well-performed and manuscript is well-written. Please explain why the TNBS model was used for the study.

>>>Responses:

We greatly appreciate the reviewer’s comments.  To make this point clearer, we added some explanations and relevant references on the background of TNBS model in the Introduction section. (Page 2, Line 79-86. In red letters)

Reviewer 3 Report

An Interesting original article describing how hydrogen water may alleviate pain sensation by suppressing inflammation in a rat IBD model, opening further experimentations also in humans about its possible  use in medicine; only minor queries: 

A description of currently available therapeutic drugs in Human IBD should be added in the introduction; here an interesting article: doi: 10.1080/03007995.2020.1786681.

Thank You and good luck!

Author Response

Reviewer 3 comments

An Interesting original article describing how hydrogen water may alleviate pain sensation by suppressing inflammation in a rat IBD model, opening further experimentations also in humans about its possible use in medicine; only minor queries: 

A description of currently available therapeutic drugs in Human IBD should be added in the introduction; here an interesting article: doi: 10.1080/03007995.2020.1786681.

>>>Responses:

We deeply appreciate the reviewer’s comments.  According to the reviewer’s comments, we added some descriptions of available therapeutic approach in IBD in the Introduction section (Page 2, Line 50-54. In red letters)